# m5CRegpred: Epitranscriptome Target Prediction of 5-Methylcytosine (m5C) Regulators Based on Sequencing Features

**DOI:** 10.3390/genes13040677

**Published:** 2022-04-12

**Authors:** Zhizhou He, Jing Xu, Haoran Shi, Shuxiang Wu

**Affiliations:** 1Key Laboratory of Ministry of Education for Gastrointestinal Cancer, School of Basic Medical Sciences, Fujian Medical University, Fuzhou 350004, China; zhe54@ucsc.edu (Z.H.); xjing955@fjmu.edu.cn (J.X.); 2Department of Molecular, Cell, and Developmental Biology, University of California, Santa Cruz, Santa Cruz, CA 95064, USA; 3Research Center for BioSystems, Land Use, and Nutrition (IFZ), Institute of Applied Microbiology, Justus-Liebig-University Giessen, Heinrich-Buff-Ring 26-32, 35392 Giessen, Germany; 4Fujian Key Laboratory of Tumor Microbiology, Department of Medical Microbiology, School of Basic Medical Sciences, Fujian Medical University, Fuzhou 350004, China

**Keywords:** 5-methylcytosine, machine learning, readers

## Abstract

5-methylcytosine (m5C) is a common post-transcriptional modification observed in a variety of RNAs. m5C has been demonstrated to be important in a variety of biological processes, including RNA structural stability and metabolism. Driven by the importance of m5C modification, many projects focused on the m5C sites prediction were reported before. To better understand the upstream and downstream regulation of m5C, we present a bioinformatics framework, m5CRegpred, to predict the substrate of m5C writer NSUN2 and m5C readers YBX1 and ALYREF for the first time. After features comparison, window lengths selection and algorism comparison on the mature mRNA model, our model achieved AUROC scores 0.869, 0.724 and 0.889 for NSUN2, YBX1 and ALYREF, respectively in an independent test. Our work suggests the substrate of m5C regulators can be distinguished and may help the research of m5C regulators in a special condition, such as substrates prediction of hyper- or hypo-expressed m5C regulators in human disease.

## 1. Introduction

Epitranscriptome is an emerging field in the past 10 years, and there are more than 170 types of RNA modifications identified [1]. 5-methylcytosine (m5C) is one of the prevalent RNA modifications, which has been found in most eukaryotes, prokaryotes, and archaea [2]. Biochemical research has revealed that m5C is abundant in tRNA and rRNA and serves a variety of molecular roles [3]. For example, m5C affects translation fidelity by altering the shape of rRNA to govern ribosome synthesis and processing [4]. The evolutionarily conserved m5C is responsible for maintaining the tertiary structures of tRNA [5]. Furthermore, new high-throughput investigations using bisulfite treatment or immunoprecipitation techniques have shown the presence of m5C on mRNA as well [6,7], which is associated with stability, export from nucleus [8], turnover [9], and translation [10] of mRNA.

Based on the developed LC–MS/MS technique, the estimated m5C/C ratio in human mRNA is about 0.02–0.09% [11]. With recent advances in genomics, at least 5 types of sequencing methods have been developed to reveal the epitranscriptome profile of m5C, including RNA-BisSeq [6], TAWO-seq [12], AZA-IP-seq [13], m5C-RIP-seq, and miCLIP-seq. These methods can be divided into two groups according to their principles: (a) chemical-dependent methods using bisulfite, peroxotungstate and 5-azacytidine in the first three methods, respectively; (b) the antibody-based methods using m5C-specific antibody and m5C-regulator antibody in the last two methods. Although the above methods provide capacity to detect m5C in transcriptome, comparing with the consensus DRACH [14] motif for m6A in most species, the exact m5C motif is still unknown. Using m5C-RIP-seq, multiple motifs for m5C were observed in *Arabidopsis thaliana*, including HACCR, CWUCUUC and CCDCCR [15], whereas m5C only showed an enrichment around CG-rich region in different species based on RNA-BisSeq [12,16,17,18].

Similar to methylation on DNA and protein, m5C is a reversible mark, which is deposited by methyltransferases and is removed by demethylases [10]. The members of NOP2/Sun RNA methyltransferase family are primary methyltransferases for m5C, including NSUN1, NSUN2, NSUN3, NSUN4, NSUN5, NSUN6, NSUN7. Some members from DNMT and TRDMT families are responsible for m5C deposition also. TET families and ALKBH1 regulated the demethylation of m5C on mRNA and tRNA which led to RNA degradation and mitochondrial activity, respectively [19]. Recent studies have reported that ALYREF and YBX1 are m5C binding proteins that can facilitate mRNA export [18] and stabilization [8] by recognizing m5C.

Although numerous effective bioinformatics studies for RNA modification sites prediction have been published in the epitranscriptomics field [20,21,22,23,24,25,26], none has focused on the substrate specificity of different m5C-related enzymes, such as methyltransferases (writers) and binding proteins (readers). In this work, we presented a bioinformatics framework “**m5CRegpred**” (which stands for m5C regulators substrate prediction, see Figure 1) based on machine learning and sequence-derived features to predict the substrate of m5C writers NSUN2 and readers YBX1 and ALFREF. The associations between diseases and m5C regulators have been reported before [27,28], especially that the hyper- or hypo-expression of NSUN2/YBX1/ALFREF were observed in multiple types of cancer [27,29,30,31,32]. This bioinformatics framework may help identify the substrate of each m5C regulators, which may provide another opportunity to understand their pathway in human diseases. The project code and training sequences are available at https://github.com/SXWuFJMU/m5CRegpred/ (accessed on 1 March 2022) and the supplement tables are available at https://github.com/SXWuFJMU/m5CRegpred/blob/main/Supplement%20Tables.zip (accessed on 1 March 2022).

## 2. Methods and Materials

### 2.1. The m5C Sites and Target Sites of the Enzymes

The transcriptome-wide m5C sites were extracted from the m6A-Atlas database [33], which were detected by four types of sequencing methods (Table 1). The sequences with 41 nt length and an m5C modification site at the center were generated to map with Par-CLIP [34] or eCLIP [35] data to identify the substrate of m5C regulators (Table 2). The substrates of m5C regulators were considered as the positive sites in the prediction. The unmethylated sites or unregulated sites from the same transcript with the positive sites were randomly selected as the negative sites, which keep the positive-to-negative ratio with 1:1. For each m5C regulator, the predictor was trained with 80% of the sites and the remaining 20% of sites were used for independent testing. To reduce the bias in the experiment, especially when selecting the polyA RNAs during library preparation, we built separate prediction models using full transcript data and mature mRNA data, respectively. In the mature mRNA predictor, only m5C sites located in exon regions are considered.

Considering the sequencing bias and non-specific binding of RNA modification antibody, the m5C sites identified from the IVT transcript [39] (in vitro transcribed RNA product consisted of only commercial NTPs, which should be free of modification) were used to filter false-positive sites and further improve the data quality. Additionally, the CD-HIT [40] software was used to remove redundant sequences with default parameters. As a result, sequence similarity is less than 85% in the dataset.

### 2.2. Sequence-Derived Features

Based on different physical and chemical properties, nucleotides can be decoded into different numeric vector or matrix. These encoding methods have been summarized in recent studies [41,42,43,44,45,46]. In this project, we selected eight popular methods in the RNA modification prediction field to identify the optimal features for substrate prediction: nucleic acid composition (CONPOSI), binary encoding method (ONE_HOT), position-specific nucleotide propensity (PSNP), electron-ion interaction pseudopotentials (EIIP), auto-correlation (autoCor), cross-correlation (crossCor), pseudo k-tupler composition (PseKNC) and chemical property (ChemProper).

### 2.3. Feature Description

**Nucleic acid composition:** Nucleic acid composition (CONPOSI) has been widely used in previous research [47]. In our study, dinucleotide frequencies were applied for sequence encoding, which can be presented as a 16-demensional feature vector (*AA*, *AC*, *…*, *UU*):$$F_{i}=\left(f_{AA},f_{Ac},f_{AG},\ldots \ldots f_{UU},\right)$$
where the f represents frequency of dinucleotide in the *i*-th sequence.

**Binary encoding method:** The nucleotide at each point in the flanking window is represented by four numeric values. The *A*, *C*, *G*, and *U* characters that fill the sequence termini were translated into binary vectors of (1,0,0,0), (0,1,0,0), (0,0,1,0), and (0,0,0,1), respectively.

**Position-specific nucleotide propensity**: The ‘position-specific nucleotide propensity based on single strand’ (PSTNPss) is a statistical method to encode the RNA sequences. In our study, the position-specific dinucleotide propensity was used, which contains 16 (i.e., 4^2^) types of dinucleotides (e.g., ‘*AA*’, ‘*AC*’, ‘*AG*’, *…*, ‘*UU*’). Therefore, for an RNA sequence with L-bp length, the dinucleotide position specificity can be formulated as a matrix, where: $z_{i,j}=F^{+}\left(diN_{i}|j\right)-F^{-}\left(diN_{i}|j\right)$
$$Z=\left[\begin{array}{ccc} Z_{1,1} & \cdots & Z_{1,L-1}\\ \vdots & \ddots & \vdots \\ Z_{16,1} & \cdots & Z_{16,L-1} \end{array}\right]$$$F^{+}\left(diN_{i}|j\right)$ and $F^{-}\left(diN_{i}|j\right)$ represent the frequencies of the *i*-th dinucleotide (diN) at the *j*-th position appearing in positive dataset and negative dataset, respectively.

**Electron-ion interaction pseudopotential:** The EIIP method was proposed by Nair and Sreenadhan [48], which considers electron-ion interaction potential values between nucleotide. The EIIP values for each nucleic acid were shown blow:$$\left\{\begin{array}{c} \begin{array}{c} A=0.1260\\ U=0.1335\\ C=0.1340\\ G=0.0806 \end{array} \end{array}\right.$$

In an RNA sequence, each nucleic acid will be replaced with its correspond EIIP value. For example, sequence ‘*GCAU*’ will be converted into a numeric vector (0.0806, 0.1340, 0.1260, 0.1335).

**Auto-covariance and cross-covariance**: the auto-covariance and cross-covariance were invented based on the physicochemical (PC) properties between two nucleotides [49]. In this work, we used ten types of PC to denote RNA, which can be formulated as a matrix:$$\mathrm{PC}=\left[\begin{array}{ccc} \begin{array}{cc} PC_{1,1} & PC_{1,2}\\ PC_{2,1} & PC_{2,2} \end{array} & \cdots & \begin{array}{c} PC_{1,10}\\ PC_{2,10} \end{array}\\ \vdots & \ddots & \vdots \\ \begin{array}{cc} PC_{L-1,1} & PC_{L-1,2} \end{array} & \cdots & PC_{16-1,10} \end{array}\right]$$
where $PC_{i,j}$ represents the *i*-th type of PC value of the *j*-th dinucleotide in the RNA sequence with L-bp length. Based on the PC matrix, the auto-covariance and cross-covariance can be calculated by following formulas, respectively:$$AC_{\lambda }^{i}=\frac{1}{L-1-\lambda }\overset{L-1-\lambda }{\underset{j=1}{\sum }}(PC_{i,j}-\bar{PC_{i}})\left(PC_{i,j+\lambda }-\bar{PC_{i}}\right)$$
where $\bar{{\mathrm{PC}}_{i}}=\frac{1}{L-1}\overset{L-1-\lambda }{\underset{j=1}{\sum }}PC_{i,j}$
$$CC_{\lambda }^{i1,i2}=\frac{1}{L-1-\lambda }\overset{L-1-\lambda }{\underset{j=1}{\sum }}(PC_{i1,j}-\bar{PC_{i}})\left(PC_{i2,j+\lambda }-\bar{PC_{i}}\right)\left(i1\neq i2\right)$$

The *AC* focuses on the correlation coefficient of the same physicochemical property between two subsequences, whereas *CC* considers the correlation coefficient between two subsequences with each belonging to a different PC property. The λ in this study equals to 39, which can capture more sequence information.

**Pseudo k-tupler composition:** PseKNC is the most widely used encoding method in the bioinformatic field, including protein, DNA and RNA prediction [50,51,52,53,54,55]. Several software/web servers/packages [41,42,43] have collected PseKNC methods in the suit. In this study, we directly used the PseKNC encoding method from ilearnplus web server to generate sequence-based features [43].

Chemical property: The sequence feature uses three unique structural chemical features to encode the nucleotide sequence: ring structures, functional groups, and hydrogen bonds. Adenine and cytosine have the amino group, whereas guanine and uracil have the keto group; adenine and guanine have two ring structures, whereas cytosine and uracil only have one; adenine and uracil can form two hydrogen bonds during hybridization, whereas guanine and cytosine can form three hydrogen bonds. Based on these chemical properties, the *i*-th nucleotide from sequence $S_{i}$ can be encoded by a vector $S_{i}=\left(x_{i},\;y_{i},z_{i}\right)$
$$x_{i}=\left\{\begin{array}{c} 1\;if\;s_{i}\in \left\{A,C\right\}\\ 0\;if\;s_{i}\in \left\{G,U\right\} \end{array}\right.\;y_{i}=\left\{\begin{array}{c} 1\;if\;s_{i}\in \left\{A,G\right\}\\ 0\;if\;s_{i}\in \left\{C,U\right\} \end{array}\;z_{i}=\left\{\begin{array}{c} 1\;if\;s_{i}\in \left\{A,U\right\}\\ 0\;if\;s_{i}\in \left\{G,C\right\} \end{array}\right.\right.$$

In other words, the *A*, *C*, *G*, *U* can be encoded as a vector (1,1,1), (0,1,0), (1,0,0) and (0,0,1), respectively.

### 2.4. Machine Learning Algorisms and Performance Evaluation

Machine learning algorithms have been widely used in many fields of biological research, such as miRNA target prediction, protein phosphorylation sites prediction, and achieved great performance in predicting RNA methylation sites. In this project, we used an R language interface of LIBSVM [56] to build Support Vector Machine (SVM) based predictors to compare encoding schemes and influence of sequence length. In addition, we compared multiple machine learning algorithms including SVM, Generalize Linear Model (GLM), Random Forest (RF), and XGBoost from R package caret [57] to identify a better algorithm for model construction. All default parameter in these functions were used to build predictors.

To validate the predictor performances, the five-fold cross-validation and independent test was employed for features selection purpose. The influences of sequence length and algorithms were evaluated by independent test only. The area under the receiver operating characteristic curve (AUROC) was calculated as the main performance evaluation metric. In addition to AUROC, the accuracy (*ACC*), sensitivity (*Sn*), and specificity (*Sp*) were calculated to measure the performance on algorithms comparison:$$Sn=\frac{TP}{TP+FN}$$
$$Sp=\frac{TN}{TN+FP}$$
$$Acc=\frac{TP+TN}{TP+FN+FP+TN}$$

## 3. Results

### 3.1. Performances Based on Different Features

Recent studies have proven sequence-derived features are high reliability and effectiveness to reflect intrinsic relation to the targets. Here, we explored and compared eight different encoding methods for predicting the target specificity of m5C-regulators. After the CD-HIT filter, there are 269, 841, and 175 sequences considered as substrates of NSUN2, YBX1, ALYREF on mature mRNA model and 335, 1137, and 381 on full transcript model.

To identify the optimal features for the m5C-regulators prediction, the performance of 5-fold cross-validation on the training data (Appendix A) and the independent test (Table 3) were both reported. In general, each feature achieved better performance on the full transcript model than the mature mRNA model, because the sequences of the exons are more conserved than sequences of introns which may have similar patterns. Among the eight encoding schemes, the PSNP methods achieved the best average performances on the regulator–substrate prediction, with AUROC scores of 0.869, 0.724, and 0.889 in independent tests of the NSUN2, YBX1, and ALYREF substrate prediction on the mature mRNA model. Although the “COMPOSITION” method had the best performances of YBX1 and ALYREF prediction on full transcript model, the performances are faint higher than PSNP method (0.764 of COMPOSITION vs. 0.763 of PSNP on YXB1 and 0.849 of COMPOSITION vs. 0.847 of PSNP on ALYREF). Additionally, the performances on mature mRNA model may reflect the actual prediction performances without overestimation due to polyA selection during library preparation [58]. Thus, the PSNP encoding method was selected to build predictors and further analysis.

Additionally, except PSNP feature, two features with lower performance were combined randomly to test their performances (Appendix A). Compared to PSNP feature, the EIIP–autoCovar combination features and EIIP–CONPOSI combination features achieved slight improvements (0.768 vs. 0.763; 0.849 vs. 0.847) for YBX1 and ALYREF substrate prediction, respectively, in full transcript model. These results suggest the PSNP may be the most appropriate feature for substrate prediction of m5C regulators.

### 3.2. Performances Based on Different Length Windows

The sequence windows length contains important sequence information and will affect the prediction performances [59,60], thus, we tried to optimize the length of the input sequences. The sequences with 21-, 31-, 41-, 51-, 61-, 71-, and 81-nt length and with an m5C modification site in the middle were tested to find the most promising prediction results (Figure 2 and Figure 3). On both the mature mRNA model and full transcript model, performances of NSUN2 and ALYREF substrate prediction were improving at the beginning, reaching the highest AUROC, and AUROC decreased as the length further increased. For YXB1 substrate prediction, the performances improved in a relatively steady manner and stabilized in the end as the length increased. Based on these results, the 51 nt and 51 nt for ALYREF, 71 nt and 61 nt for NSUN2, and 71 nt and 61 nt for YBX1 were selected on the mature mRNA model and full transcript model, respectively. These selected sequences can be freely accessed at https://github.com/SXWuFJMU/m5CRegpred/ (accessed on 1 March 2022).

### 3.3. Performances Based on Different Machine Learning Algorithms

Although the SVM is the most popular algorithm on the RNA modification prediction filed [25,58,61,62,63,64,65,66,67], we also conducted a system comparison for the performances among SVM, RF, GLM, and XGBoost. The AUROC, accuracy, sensitivity, and specificity were calculated to measure the performance of predictors by independent test (Figure 4). In general, the performances were stable when the optimized sequence lengths were used among different machine learning algorithms, despite the SVM has the best performances.

### 3.4. Data Interpretation

Although the exact motif for m5C modification sites is unknown, the motif of YBX1 substrate was identified as “CA(U/C)C” in human [68] before YBX1 is considered as an m5C reader. Further, Chen et al. [8] and Yang et al. [69] proved that YBX1 preferred to bind with “CA(U/C)m5C” rather than unmethylated “CA(U/C)C”, which suggests “CA(U/C)C” may be one motif for YBX1 dependent m5C. However, the motif of ALYREF is unclear. Another study based on NSUN2 knockout suggests “NGGG” is enriched among the NSUN2-dependent m5C sites [16]. In this project, after the YBX1 CLIP data mapping with m5C sites, there are 53 (3.36%, 53/1576) sequences contained “CA(U/C)C” motif. For NSUN2 data, there are 227 (55.6%, 227/408) sequences containing “NGGG”. The modification sites with potential motif were summarized in Appendix A Appendix A.

To better understand which sequences may contribute to the predictors, the motif among training data were analyzed by the STREME [70] from MEME suit. The most enriched motif for each regulator was presented in Figure 5. The results are similar to the previous studies, the GC-enriched sequences are around the m5C sites, regardless of whether there are substrates of NSUN2, YBX1, and ALYREF. Additionally, the motif for the YBX1 motif is insignificant, which may explain the lower performance of substrate prediction of YBX1 and suggest extra sequence-based features can be considered for the performance improvement. We also analyzed the motifs of the false-negative sites in the independent test. The false negative data of NSUN2 substrate are enriched in the GA-enriched sequences, whereas motifs for ALYREF false negative sites were different in full transcript and mature mRNA model.

### 3.5. Case Study

The low resolution m5C profile on the breast epithelium cell line MCF10A generated by the m5C-meRIP technique was obtained from GSE53370 [71]. There are 1,744,029 cytosines located on the m5C peaks, and each cytosine was considered as the putative methylation sites. After prediction by m5CRegpred, 16,313 cytosines (Appendix A) were considered regulated by NSUN2 and recognized by at least one m5C reader with high confidence (probability > 95%). Among these results, cytosine located in gene PTPN2 (chromosome 18: 12789928), which is a tumor suppressor gene [72] with a low expression ratio in breast cancer [73], was a putative site regulated by NSUN2 and YBX1. The hypo-expression of NSUN2 were observed in the breast cancer [30], which may cause the low m5C level on mRNA. Although gene expression level of YBX1 is undifferentiated [74], less PTPN2 mRNA will be recognized by YBX1 due to the low methylation. Considering the YBX1 can stabilize mRNA [8], the impaired recognition by YBX1 will lead to the decay of PTPN2 and contribute to the development of breast cancer.

## 4. Discussion

In the past 10 years, RNA modifications-associated biological processes and molecular functions were widely explored to suggest the epi-transcriptome is an important layer in epigenetics regulation. The function and disease association of m5C were discussed also. Although the importance of m5C was proven, the attention on m5C modification is still less enthusiastic than m6A modification due to the lack of a dependable detecting method [7,16,60]. Here, we presented a bioinformatics work to show the substrates of m5C regulators can be distinguished by machine learning approaches, which provide a convenient and fast way on m5C relevant studies. In this study, we compared different encoding methods, length windows, and machine learning algorithms to build the optimal predictor (AUROC scores 0.869, 0.724, and 0.889 for NSUN2, YBX1, and ALYREF, respectively) on mature mRNA model. However, there are some limitations in the current study. The major defect is the bias of prediction results. The bias of result was considered in the site prediction field, such as using likelihood ratio (LR) to justify the probability. In these studies, the likelihood ratio was calculated by the probability of motif occurrence and the probability of observed RNA modification. Considering the motif of m5C is unclear (only can be summarized as the GC-enrich region) and the probability of observed RNA modification cannot be replaced with the probability of m5C regulated by NSUN2/YBX1/ALYREF, the bias is difficult to be calculated based on current knowledge.

Additionally, there are some shortcomings can be improved in further study. Firstly, although the sequence-derived features-based predictors have achieved acceptable performances, the advanced genomic features [75] should be considered to improve the performance in the future, especially for YBX1 substrate prediction. Secondly, the deep learning algorithms which were applied in the site prediction studies [76,77,78,79] recently, have better power than machine learning. Therefore, deep learning can be used to improve performance. Thirdly, the current prediction only focuses on one methyltransferase and two readers due to limited published dataset. More m5C regulators will be considered further once the sequencing results are released. Finally, some recent studies have suggested that RNA modification regulation is tissue-specific. Thus, the elaborate prediction with the tissues/cell lines specific should be considered in further research.

## Figures and Tables

**Figure 1 genes-13-00677-f001:**
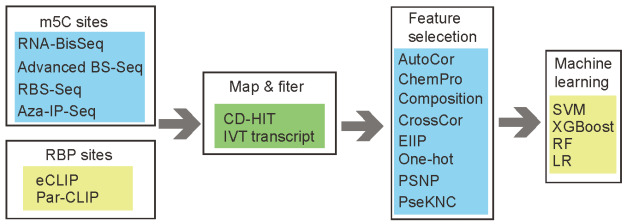
**The workflow for m5CRegpred.** The methylation sites and RNA Binding proteins (RBP) sites were obtained from four and two types of sequencing techniques, respectively. Eight kinds of encoding methods were considered in the project.

**Figure 2 genes-13-00677-f002:**
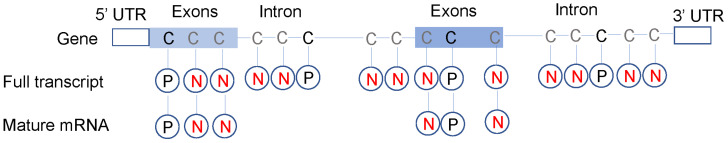
**Full transcript model and mature mRNA model.** To select negative sites, the unmodified sites and methylated sites un-regulated by NSUN2/YBX1/ALYREF from the intron and exons were both considered in the full transcript model; whereas the mature mRNA model only considered sites from exons. As most captured sequences during library preparation are exons (mature mRNA) due to polyA selection, the performance of full transcript model will be overestimated.

**Figure 3 genes-13-00677-f003:**
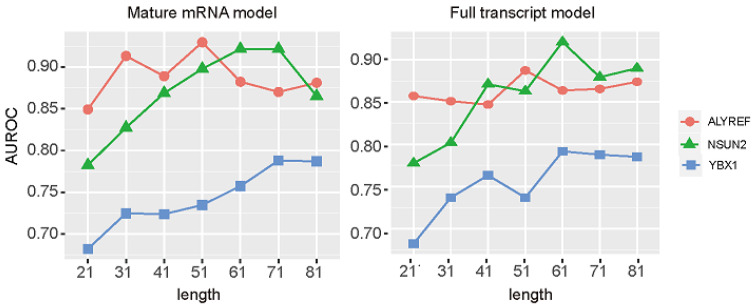
Performance of different length windows with PSNP encoding method.

**Figure 4 genes-13-00677-f004:**
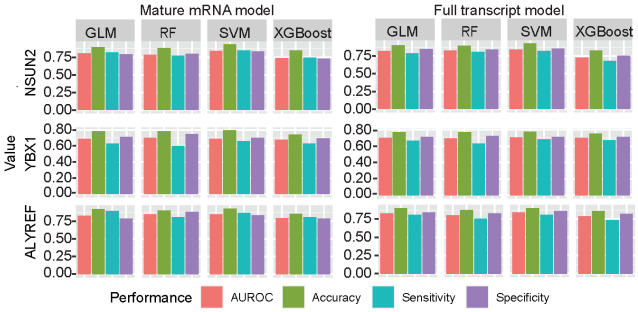
**Performance analysis on different machine learning algorithms**. SVM represents for support vector machine, RF represents for random forest and GLMs represent for generalize linear model.

**Figure 5 genes-13-00677-f005:**
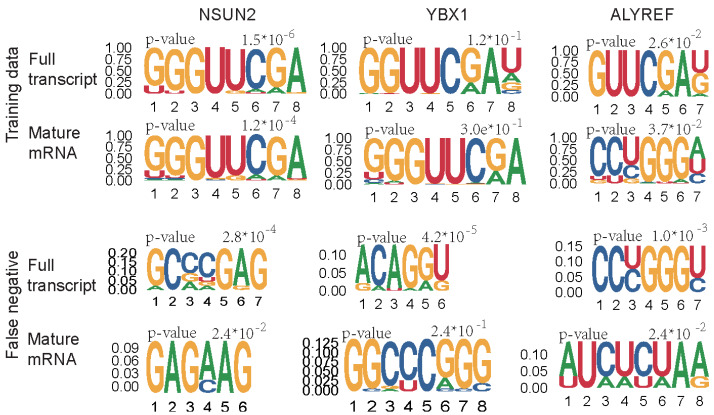
**Motif discovery for the training data and****false-negative sites of independent test data****.** For the training data, only the positive sites were used for motif discovery. The motif with width of 4 bp to 8 bp was scanned by STREME. The different motifs of ALYREF in training data may be due to the different data size, which contained 296 sequences in the full transcript model whereas only 137 in mature mRNA model.

**Table 1 genes-13-00677-t001:** Base-resolution datasets of m5C sites.

ID	Technique	Source	Cell Line	Ref.
1	RNA-BisSeq	GSE93751	HeLa	[18]
2	RNA-BisSeq	GSE133671	T24	[8]
3	BS-seq with improved protocol	GSE122260	HEK293T	[16]
4	BS-seq with improved protocol	GSE122260	HeLa	[16]
5	RBS-Seq	GSE90963	HeLa	[36]
6	Aza-IP	GSE38957	HeLa	[37]

**Table 2 genes-13-00677-t002:** Target sites of m5C regulators identified by Par-CLIP or eCLIP.

	Protein	Cell Line	Technique	Source	Ref.
Writer	NSUN2	K562	eCLIP	GENCODE	[38]
Reader	YBX1	T24	PAR-CLIP	GSE133620	[8]
	ALYREF	T24	PAR-CLIP	GSE133620	[8]

**Table 3 genes-13-00677-t003:** Independent test with different features.

	Mature mRNA Model			Full Transcript Model			Average
	NSUN2	YBX1	ALYREF	NSUN2	YBX1	ALYREF	
EIIP	0.656	0.656	0.807	0.721	0.764	0.849	0.742
autoCor	0.567	0.546	0.584	0.523	0.617	0.710	0.591
crossCor	0.594	0.520	0.718	0.609	0.597	0.679	0.620
PseKNC	0.660	0.622	0.723	0.738	0.732	0.774	0.708
ChemProper	0.602	0.649	0.665	0.698	0.692	0.778	0.681
ONE_HOT	0.606	0.646	0.668	0.708	0.690	0.778	0.683
CONPOSI	0.656	0.656	0.807	0.721	0.764	0.849	0.742
PSNP	0.869	0.724	0.889	0.871	0.763	0.847	0.827

## Data Availability

All relevant data is provided in the manuscript. Please contact at wushuxiang@fjmu.edu.cn for any raw data files and further analysis.

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
