# Peer review of "m5CRegpred: Epitranscriptome Target Prediction of 5-Methylcytosine (m5C) Regulators Based on Sequencing Features"

_genes, 2022, doi:10.3390/genes13040677_

Round 1

Reviewer 1 Report

5-methylcytosine is a common and largely studied post-transcriptional modification consisting of cytosine methylation. This reaction is known to be reversible, and it is mediated and regulated by several m5C-related enzymes (writers) and recognized by a variety of binding proteins (readers). Developing tools able to predict if a given sequence could represent a putative binding site for m5C related writers (and/or readers) is a very interesting topic.

In this context, Zhizhou He et al., focused their work on building a new bioinformatics tool, m5CRpred, based on machine learning supervised approaches to address this need. The final goal was to predict, from nucleotide sequences, putative candidate substrates for m5C binding proteins. 

In general, the aim and proposed approaches seem to be very promising. However, the paper is too short and many methodological aspects are not deeply explained. This is quite important for reproducibility purposes.

Major concerns:

  • The lack of supplementary materials does not help final users in understanding methodological details.
  • It could be useful to better explain the filtering step made by IVT transcript and CD-HIT software.
  • Widely describe, discuss and argue the choice of the 8 different algorithms used for encoding sequences fed to machine learning predictors.
  • Have the hyperparameters been tuned during cross-validation and training phases?
  • In figure 3, are the shown performances related to CV validation sets? It is not clear.
  • Figure captions (e.g., algorithm/features encoding used) should be considered to improve the readability of the work.

In addition:

is there a GitHub repository for m5CRpred? Is the code available? Is there a manual?

How do researchers explain the lower performances of PSNP encoding method on YBX1 shown in figure 2?

Which are the characteristics of wrong classified sites and thus in turn, which bias (if present) is expected when using m5CRpred?

Also, a wider exploratory data analysis of the input data should be performed prior to models training, in order to visualize graphically the whole dataset.

Finally, did researchers schedule to use all these trained weaker predictors to build an ensemble version to further improve performance metrics or to combine into a unique classifier all the encoded sequences? Each encoding strategy may have led to different information loss and transmission toward classification procedures, and this could change final performances.

Reviewer 2 Report

Comments on “ m5CRpred: epitranscriptome target prediction of 5-methylcyto-sine (m5C) regulators based on sequencing features.” by He et al.

  1. The abstract should mention what are NSUN2, YBX1 and ALYREF.
  2. The last sentence of the abstract mentions “special conditions”. Authors should elaborate in brief what are these special conditions.
  3. The introduction talks about the RNA methyltransferases but there is no mention of methylation site. Are all RNA methyltransferases methylate CpG or cytosine in some other consensus sequence? Since the manuscript is on the methylation site, authors should devout a paragraph on the details of differences in the cytosine methylation sites. Same is true for the proteins that bind to the methylated cytosines containing sequences. Is any preferred sequence known?
  4. What is the primary methylation site of NSUN2 and primary identification site of YBX1 and ALYREF?
  5. The authors term the bioinformatics framework “m5CRpred”, however, “m5CRpred-SVM” is already used by Chen et al (Ref. 15).
  6. Authors should mention association between diseases and 5mC by giving some concrete examples. The introduction is vague and does not emphasize why this study is important.
  7. Authors mention supplementary material; however, none is available for review. Without the supplementary material, evaluation of research design is not possible. Please provide supplementary material along with the manuscript.
  8. For the benefit of the reader, authors should elaborate Par-CLIP and eCLIP with citation of appropriate references.
  9. There is a problem with the sentence “In general, each features archived better performance on the full transcript model than the mature mRNA model, because ……”. What is the difference between conservation and similar pattern? If exons have more conservation and introns have similar patterns, why there should be a difference between the performance on full transcript versus mature RNA?
  10. Authors should provide the consensus sites of NSUN2, YBX1 and ALYREF in the results rather than stashing them in the supplementary material.
  11. Thus, at the end of the results, authors have used only 1 methylase and 2 methylated RNA binding proteins. To claim that “we presented a bioinformatics work to show the substrates of m5C regulators can be distinguished by machine learning approaches, which provide a convenient and fast way on m5C relevant studies.” authors should analyze sites for more RNA mathylases.
  12. Discussion should bring in biological relevance of the work. If the site of methylation is known (say 101 bp with methylated cytosine in the middle), is it possible for the authors to predict which methyltransfetase has methylated it, which binding proteins will bind it and lead to a specific disease.
  13. The manuscript needs language editing and spelling correction at several places.

Round 2

Reviewer 1 Report

The manuscript has been improved and the authors answered all the raised questions. However, a further concern has to be taken into account before publication. Indeed, it is not easy how to use the m5CRpred framework in practice. Although a Github repository has been provided and shared, the documentation is too limited and must be improved. Also, at least a practical example (including comments) has to be provided.

Reviewer 2 Report

Comments on the revised version of “m5CRegpred: epitranscriptome target prediction of 5-methylcytosine (m5C) regulators based on sequencing features” by He et al.
Responses to Q.4 and Q.10 are not satisfactory. It gives an impression that the authors have put forth big theories about which 5mC sites regulated by m5C writer NSUN2 and m5C readers YBX1 and ALYREF without even knowing the consensus sites or motif of cytosine methylation. The point needs to be satisfactorily answered.
Figure 5: Why motif in the full transcript is different from motif in the mature RNA for ALYREF, in the training data?
Case study: PTPN2 has 20 m5Cs, 19 bound by ALYREF and 1 bound by ALYREF as well as YBX1. Do authors mean that the binding (and regulation) by only YBX1 to 1 m5C in PTPN2 would contribute to the development of breast cancer? In that case, what is the role of ALYREF?
Discussion: In the discussion, authors should discuss the limitations of the current study rather than what can be addressed in the further study.
Please check grammar in the sentences:
However, there are some limitations can be addressed in further study.
These selected sequences are freely accessed at ………………
Supplementary Table 3: Please indicate what “yes” and “-” mean.
